# Weight Gain and Asthenia Following Thyroidectomy: Current Knowledge from Literature Review

**DOI:** 10.3390/jcm11185486

**Published:** 2022-09-19

**Authors:** Gregorio Scerrino, Giuseppe Salamone, Alessandro Corigliano, Pierina Richiusa, Maria Pia Proclamà, Stefano Radellini, Gianfranco Cocorullo, Giuseppina Orlando, Giuseppina Melfa, Nunzia Cinzia Paladino

**Affiliations:** 1Unit of General and Emergency Surgery, Department of Surgical Oncology and Oral Sciences, University of Palermo, Via L. Giuffré, 5, 90127 Palermo, Italy; 2Department of Health Promotion Sciences Maternal and Infantile Care, Internal Medicine and Medical Specialties (PROMISE), Section of Endocrinology, University of Palermo, Via del Vespro 129, 90127 Palermo, Italy; 3Department of General Endocrine and Metabolic Surgery, Conception University Hospital, Aix-Marseille University, 147 Boulevard Baille, 13005 Marseille, France

**Keywords:** thyroidectomy, weight gain, chronic asthenia, BMI, metabolic syndrome

## Abstract

Background: Thyroidectomy is a common procedure all over the world. Its complication rate is low, but some patients complain of weight gain and/or asthenia. The aim of this review is to investigate the correlation between thyroidectomy and weight change and asthenia. Materials and Methods: Seven papers concerning weight gain and four concerning asthenia were found. Results: Weight gain would seem to be more related to the change in habits after surgery. Asthenia seems to be more linked to endocrine mechanisms not yet clarified although a deficiency of triiodothyronine and its metabolites could explain some of its aspects. Conclusion: Patients who are candidates for thyroidectomy should be adequately informed of the onset of both possible implications of the surgical act in terms of weight gain and chronic asthenia.

## 1. Introduction

Thyroidectomy is a common practice in several countries [1,2]. This procedure is generally considered safe. However, several scientific societies and individual working groups have developed specific informed consent forms that alert patients to complications of the procedure, their frequency, and pathways to be implemented for their solution [2,3]. Hyperthyroid patients that undergo thyroidectomy experience weight loss due to reduced circulating thyroid hormones [4]. Post-thyroidectomy hypothyroidism could be the cause of both weight gain and fatigue [5] although, even in formally euthyroid patients, the increase in body mass index (BMI) and the onset of chronic asthenia have been described [4,6]. A population-based study focused on weight changes following thyroidectomy has already been performed. A systematic review with meta-analysis of data was added to this study [4]. It included some studies focused on the effects of subclinical hyperthyroidism on weight loss and patients under TSH-suppressive treatment. The present review is aimed at assessing once again whether there is a correlation between thyroidectomy and weight gain and/or asthenia in light of the clearer and updated knowledge. If so, the secondary aim is to assess whether there are common pathogenic steps shared by the two sequelae. From this perspective, it will be possible to evaluate which possible treatments for each or both sequelae to investigate in the future.

## 2. Materials and Methods

The research of bibliographic sources was performed with the PRISMA criteria. The search process was reported in a provided flow chart (Figure 1).

The MESH terms “thyroidectomy”, “weight gain”, and “asthenia” were searched for in *PubMed*, *Web of Science*, and *Scopus*. Since the combination of all three words resulted in only one article not relevant to the purpose of the search, we searched with the two pairs of MESH terms separately, i.e., “thyroidectomy” and “weight gain”, and then “thyroidectomy” and “asthenia”. Only articles in English from 2010 to 2022 were considered. Reviews were not included. When we found some articles focusing on the relationship between thyroidectomy and weight gain or asthenia, we carefully read, analyzed, and included them when the data obtained matched the goal of this study.

Any discrepancies were discussed, and if needed, a third researcher evaluated that paper.

The titles of 20 papers focused on asthenia and 31 concerning weight gain were initially retrieved. If they were in accordance with the aim of review, each abstract was evaluated. 

We retrieved 10 titles concerning weight gain and 4 on asthenia. Two abstracts from the first group were determined to be not in accordance with the aim of the search, and one more paper was excluded after reading. All abstracts from the second group were considered in agreement with the search criteria. We did not find any secondary pertinent references. 

The level of evidence of studies included were evaluated according to Sackett’s classification [7]: a recommendation supported by meta-analysis or large randomized trials (clear cut-off, low risk for error) is considered as Level I of evidence; if supported by small randomized trials and moderate/high error risk, it is classified as Level II of evidence; when supported by non-randomized, prospective with contemporaneous control trials, it is classified as Level III of evidence; studies classified as level IV are non-randomized, with historical controls or retrospective analysis; and level V concerns case series without controls and expert opinion. PRISMA criteria were applied for carrying out the search process [8]. 

The search was conducted by four independent authors who collected manuscripts independently. Four additional authors derived data from the studies that were broken down by groups. Data collected included: author, year, benchmark of each article, methods, characteristics of the control group, and pathogenic hypotheses. 

The manuscript was drafted and revised by the authors responsible for the study’s conception and design. 

We only performed a qualitative synthesis of the data obtained. Because of the scarcity and non-homogeneity of data obtained from the selected papers, meta-analysis was not performed.

## 3. Results

The search process used to carry out the present study was performed as shown in Figure 1.

The articles concerning weight gain selected for this literature review all referred to retrospective studies except for one, which was a retrospective study of prospectively collected data. The articles concerning chronic asthenia were one retrospective, one retrospective carried on prospectively collected data, and two prospective. Two of these articles were carried out at our institution. A total of seven studies concerning weight gain and four studies focused on chronic asthenia were considered eligible for this review. None of the studies dealt with the two topics from a unified perspective.

### 3.1. Weight Gain

A retrospective study by Jonklaas (2011) evaluated 120 patients who had undergone thyroidectomy for benign non-functioning goiter. These patients were normal-weight before surgery and euthyroid under replacement therapy by determining whether they would gain weight even in the absence of a postoperative transient hypothyroidism.

The authors created three age- and gender-matched control groups of 120 people each.

The first was composed of hypothyroid patients under substitutive treatment for Hashimoto’s thyroiditis. The second group of patients suffering from thyroid carcinoma was on suppressive therapy with levothyroxine (L-T4). The third one was of healthy subjects. The study investigated three questions: (i) Do the weight gain and increase in BMI differ between thyroidectomized and hypothyroid patients? (ii) Is the weight gain and BMI increase of both groups different from the annual weight gain of the population and comparable in demographics? Moreover, is it different from the weight gain of the two control groups (euthyroid subjects and patients with thyroid cancer)? (iii) Is the weight gain different for women of childbearing age or men in either the thyroidectomy or hypothyroid groups? The results were evaluated over a period of five years of observation. The study showed a weight gain and BMI increase in 86% of thyroidectomized patients and in 70% in hypothyroid patients. The weight gain and BMI increase were significantly greater in the thyroidectomy and hypothyroid groups compared to the estimated increase in a similar population and were more evident in menopausal females. Overall, the study showed that the group of patients most exposed to weight gain were hypothyroid patients, with thyroidectomies in first place and those with Hashimoto’s thyroiditis in second place although these patients remained euthyroid from the early stages of evaluation (for thyroidectomized patients, from the first moments after surgery). The authors did not clarify the possible causes of the increase in weight that was higher than expected. The pathogenetic hypotheses formulated by the authors [9] were stress of surgery, failure of L-T4 in performing its functions beyond the formally correct endocrine balance with regard to balancing the metabolic and orexigenic actions, triiodothyronine peripheral deficiency, and other contributing physiological or psychosocial factors.

Weinreb (2011) performed a retrospective comparison of the data of 102 patients operated on for thyroid cancer and 92 patients in follow-up for goiter or thyroid nodules. The authors measured weight and BMI changes over a long period of time and did not find significant differences in terms of weight gain, BMI, and BMI percentile in each group even after gender subdivision. No difference was even found between patients with suppressed TSH and those with TSH in the normal range. If the patients with TSH suppression were excluded, the data did not even change. The authors concluded by affirming that adequate doses of thyroxine allow one to maintain a normal weight and BMI [10].

Rotondi (2014) evaluated the effects of thyroidectomy on body weight changes in a cohort of 267 patients suffering from euthyroid diseases (benign goiter or malignancy) or hyperthyroidism (Graves’ disease or nodular toxic goiter). Any condition other than thyroid function that could lead to possible interference with changes in body weight was excluded. A significant increase in body weight was detected 9 months after thyroidectomy for the entire study group; the same result was observed by separating males from females. Overall, there was an increase in weight in over 58% of the subjects included in the study and a reduction in about 22%, while in 19.5%, the weight was stable in the two periods. The percentage of body weight changes in body weight was similar in the different thyroid diseases included. Applying a multiple regression model, in which several covariates were studied, the only one that showed statistical significance was the ∆ body weight detected 40–60 days after thyroidectomy. In particular, neither the extent of the thyroidectomy, TSH preoperative values, thyroid disease, nor demographic data influenced weight variation in any way. This study also formulated some pathogenetic hypotheses, clarifying first of all that weight gain in hypothyroidism is probably caused by increased body water and speculating on the possible role of leptin in the regulation of energy homeostasis, promotion of thyrotropin-releasing hormone gene expression, and conversion of T4 to T3 performed by deiodinases. An important indicator of the results of this study is the close correlation between early weight gain and long-term outcomes. This may suggest an evaluation of lifestyles induced by the intervention, such as sleep–wake rhythm, eating habits, and early recovery of physical activity [11].

In order to evaluate the weight changes in a cohort of patients that underwent thyroidectomy for benign non-toxic disease, Lang (2016) compared 898 patients suffering from benign non-toxic goiter (study group) with two control groups with similar baseline characteristics: (ii) 179 patients with benign non-toxic goiter under follow-up alone and (iii) 80 patients who had undergone excision of a single parathyroid adenoma. By establishing a baseline consisting of the two surgical groups (i) and iii)) from the perioperative period (1–2 weeks before surgery) and the non-surgical group (ii) from the first outpatient examination for thyroid disease, weight measurements were taken in the previous 12 months and in the following 6 and 12 months. As a result, during the preceding 12 months, the study group showed a significant weight gain. On the contrary, group ii showed a significant weight loss, and group iii did not show a weight change. In the post-baseline period, all groups increased in weight, but only the study group showed a mean increase of 0.9 kg compared to the 0.3 kg of both control groups. In the following 12 months, the weight increase, expressed as a percentage of baseline, was 2.6 ± 4.2%, 0.6 ± 2.4%, and 0.4 ± 1.8%, respectively, for the study group and the (i) and (ii) control groups. The percentage of patients gaining weight during this period also varied significantly: 76.8% vs. 60.9% and 55%, respectively, for the study group and (i) and (ii) control groups (*p* < 0.001). The values of TSH and LT4 showed a decrease and an increase, respectively, but remained well within normal ranges. In a regression analysis, the extent of excision did not influence the weight gain; on the contrary, younger age and higher TSH values at baseline appeared significantly associated with weight gain at post 12 months. Weight gain has been attributed to changes in dietary habits, lifestyle, and/or body metabolism [12].

A retrospective study of Glick (2018) selected 79 patients operated on for benign disease (hyperfunctioning or euthyroid goiter), and all underwent total thyroidectomy in an observation period of 18 months. The patients were divided into two groups: those whose weight gain exceeded 2% and those who had a lower gain; the latter were further divided between those who lost weight and those who kept their weight stable. After a median period of 18 months, 25 patients (32%) lost 2% or more of their baseline weight, 26 patients (33%) were stable, and 28 (35%) gained > 2% of their baseline weight.

A Student’s *t*-test revealed no significant difference between pre- and postoperative measurements in the overall group. Moreover, there was no correlation between thyroid function nor TSH levels and body weight change post thyroidectomy. The study allows us to affirm that the weight gain involves a minority of patients [13].

A population-based study performed by Ospina (2018) included 435 patients from an epidemiological registry established in Minnesota, the Rochester Epidemiology Project (REP). The patients enrolled in the study were divided into three groups: 181 patients were thyroidectomized because of thyroid cancer; 226 patients were followed for benign thyroid nodules; 28 patients had undergone thyroidectomy for benign disease. Total thyroidectomy was performed in 83% of malignancies and in 25% of benign diseases. Radioiodine ablation was performed in 30% of cancers. The observations of this study can be summarized as a slight weight gain in all the groups examined. This can be explained mainly by advancing age during the observation period. The same study included a systematic review with meta-analysis of the data available up to that point, the results of which will be examined in the Section 4 [4].

Park (2019) investigated the relationship between thyroidectomy and obesity. In the same study, the role of the extent of thyroidectomy was also evaluated. A total of 227 patients were analyzed, 38.8% of whom were already obese at the time of surgery (definition of obesity: BMI ≥ 25.0 kg/m^2^). This percentage became 39.6% in the follow-up ranging between 6 and 60 months. The conclusion of this study was not in agreement with a relationship between thyroidectomy and obesity. Additionally, heavy alcohol consumption and preoperative obesity were significant risk factor for postoperative weight gain at a multivariate analysis. The extent of thyroidectomy showed no significant influence on weight gain [14].

### 3.2. Chronic Asthenia

Focusing on post-thyroidectomy chronic asthenia, four studies were found with our search criteria.

Rosato (2014) first described this syndrome, which is commonly reported but without any scientific evidence so far. Two groups of patients evaluated retrospectively of 100 subjects each were compared; one underwent total thyroidectomy, and the other underwent thyroid lobectomy. The objective of this study was three-fold: to evaluate the presence of asthenia in patients undergoing total thyroidectomy, to assess whether thyroid removal can cause the syndrome, and finally, to reconsider the indication for total thyroidectomy. The two groups were homogeneous in demographic characteristics and free of comorbidities and post-operative complications, such as laryngeal nerve palsy or hypoparathyroidism. Anxiety and depression were highlighted with an appropriate questionnaire. Residual thyroid tissue, bilateral or at the surgical site in the case of previous lobectomy, was ruled out by ultrasound 1 year after surgery. An average of 25% of patients who had undergone total thyroidectomy showed signs of asthenia, as measured by a specific questionnaire, the brief fatigue inventory (BFI). No patient who underwent lobectomy showed such a sequela. Moreover, a univariate analysis was performed in the group of total thyroidectomy that showed no significant differences in age, gender, and BMI related to asthenia. The conclusions of the study were in favor of the real impact of thyroidectomy on postoperative chronic asthenia as a sequela of the thyroid ablation; it would be appropriate to test the effectiveness of the combination of L-Thyroxine and Tri-iodothyronin as substitutive treatment; finally, total thyroidectomy should be performed only when strictly indispensable [6]. The article by Scerrino (2017) evaluated prospectively the real impact of postoperative chronic asthenia in a cohort of 231 patients who had undergone total thyroidectomy or lobectomy and was observed from the moment of surgery to the following 12 months. As covariates, specific habits (smoking, alcohol), chronic diseases (cardiac, pulmonary, hepatic, renal, and diabetes mellitus) in no severe stage, and mental disorders (anxiety or depression) were also evaluated. Thyroid malignancies, Graves’ disease, variations in hormones outside normal limits, and an increase in BMI beyond the normality threshold were excluded from the study. Anxiety and depression were stable during the postoperative period of observation. Asthenia increased significantly in the group of total thyroidectomy. The covariates analyzed (chronic diseases, habits) did not influence the trends of chronic asthenia. The TSH values remained stable throughout the observation period, and there were no changes depending on the type of intervention performed. An imbalance of an unknown hormone produced in the gland was invoked as a possible cause of chronic asthenia. In particular, a possible insufficiency of “non-classical” hormones such as 3′,5′-diiodothyronine (T2), possibly linked to the variability of the deiodinase pattern, could explain in some aspects why some patients undergoing total thyroidectomy suffer from chronic asthenia. It can be concluded that chronic asthenia is an entity that really impacts on patients undergoing total thyroidectomy, is not present in subjects undergoing lobectomy, is not related to other chronic diseases with which the patient may be affected, and finally, the recommendation to perform total thyroidectomy surgery only when strictly necessary is confirmed [15]. Luddy et al. evaluated the post-operative health related quality of life and risk of asthenia in patients who had undergone thyroidectomy [16]. Patients were divided into four groups, namely benign disease + hemithyroidectomy, benign disease + total thyroidectomy, malignancy + hemithyroidectomy, and malignancy + total thyroidectomy, and surveyed at four time points: baseline, 2 weeks, 6 months, and 12 months after surgery. Asthenia ranged from 3% in patients undergoing hemithyroidectomy for benign disease to 48% in patients undergoing total thyroidectomy for malignancy.

In the article by Scerrino et al., 233 patients undergoing total thyroidectomy were compared with 43 subjects undergoing removal of one or two parathyroids for primary hyperparathyroidism and 43 patients undergoing laparoscopic cholecystectomy for uncomplicated cholelithiasis [17]. To date, this is the first study in which the course of asthenia in patients undergoing thyroidectomy was compared with that of patients undergoing other types of surgery. All patients enrolled in the study had a BMI in the normal range for the Caucasian race, no thyroid autoimmunity, and were euthyroid. Furthermore, in this study asthenia was assessed with the brief fatigue inventory. Patients were evaluated in three periods: preoperatively, 6 months after surgery, and 12 months after surgery. The study showed that in the thyroidectomy group, asthenia occurred in the control group at 6 months after surgery and remained almost constant in the second control group. In the parathyroidectomy group, the asthenia score was on average higher at baseline and declined significantly at follow-up, as is to be expected given the characteristics of the disease in question. Finally, in the cholecystectomy group, there was no significant variation in asthenia between the three periods examined.

One finding of this study is that the onset of asthenia seems to be favored by kidney damage, whereas male sex seems to be a protective factor.

To explain the onset of asthenia in subjects who had undergone a total thyroidectomy, the authors mention a possible imbalance in “non-classical” thyroid hormones such as T2, whose activity is particularly intense in the mitochondria; polymorphism of the enzymes responsible for deiodination, especially deiodinase 3; and a lack of melatonin, of which the thyroid is the second source of production.

## 4. Discussion

Several studies [18,19,20] looked into the link between endocrine surgery and quality of life (QoL).

Aside from the actual complications, thyroidectomy may be burdened by sequelae that, while not causing real morbidity or risk to life, can affect QoL and thus require adequate information [21,22]. There is conflicting evidence about the effect of thyroidectomy on weight change. On the contrary, the observations regarding the relationship between chronic asthenia and thyroidectomy are unequivocal in considering this correlation close and exclusively limited to total thyroidectomy. These observations are, however, limited to four articles, each developed with different methods and types of control groups and with different patient selections. Table 1 and Table 2 summarize, respectively, the articles on weight gain and on asthenia, with the data derived from the articles included in this review. As in Table 1, three articles show a relationship between thyroidectomy and weight gain, while the other three tend to exclude it.

The article by Park [14] highlighted a correlation between obesity and thyroidectomy only in patients with specific risk factors, such as preoperative high BMI and high alcohol consumption. In order to define a trend of the literature on this subject, it should be stressed that the study of Lang (2016) is based on a very extensive study group evaluated on prospectively collected data [12]. All studies included have significant limitations and biases. Moreover, it should be noted that in many of them, it is not clear whether patients undergoing total thyroidectomy or more conservative surgery were enrolled. The only three articles dealing with the problem agreed that the extension of thyroidectomy has no influence on body weight gain [11,12,14]. In this light, it would seem unlikely that the possible correlation between thyroidectomy and body weight gain could be related to endocrine mechanisms but rather to a change in habits or to some psychic factor somehow related to the intervention. In fact, in more than one article, factors related to changes in lifestyle or eating habits, sleep–wake rhythm, physical activity, or other psychological and psychosocial factors have been called into question [9,11,12]. However, the research is still far from having found a convincing cause for this association even though the research admits its real existence. In any case, the pathogenetic hypotheses formulated in some articles are purely speculative, as they are not supported by specific experimental data.

A more recent systematic review with meta-analysis concludes that thyroidectomy is associated with mild weight gain, which is particularly evident in younger patients, especially those with hyperthyroidism. These results are worth considering despite the considerable heterogeneity of the trials included in the study [23]. Furthermore, this systematic review highlights the possible implications of weight gain with diabetes and, consequently, with health status.

Contrary to what has been observed about weight gain, the literature on the association between chronic asthenia and thyroidectomy, although scarce in number of articles, is concordant in at least two aspects: the onset of chronic asthenia syndrome in a number of thyroidectomies and its exclusive association with total thyroidectomy. This syndrome is described in a percentage ranging from 25 [6] to over 49% [16]. Among the patients who underwent thyroid lobectomy, one patient in the first two studies [6,15], which can be considered as absolutely anecdotal, and in 3–6% of thyroid lobectomies for benign and malignant pathology, respectively, showed an increase in the level of asthenia found through the questionnaire adopted, which was the same in the four studies. These data seem to lead to a distinction between the two thyroidectomy sequelae. Asthenia seems to be more related to endocrinology factors not yet well-known. In fact, the hypothesis of the role of “nonclassical” thyroid hormones may appear suggestive. These hormones are produced by different biochemical mechanisms and in particular by deiodination in peripheral tissues and the thyroid gland itself [15,24]. More precisely, T2 has a strong activity in promoting metabolism at the mitochondrial level [25]. In an animal model, 12 h after T3 administration, the T2 serum concentration is increased [26]. It has also been shown that T2 promotes a rapid response of cytoplasmic organelles to changes in energy demand. It should be due to the action of T2 in specific mitochondrial pathways [27]. Although the mechanisms involved in the production and clearance of T2 are not yet well-known, the steps leading from T3 to T2 may be less effective, at least in some patients deprived of the thyroid gland, even if treated with L-T4 [15,28]. Having established that T2 could play an important role in energy metabolism at the mitochondrial level, particularly in some peripheral tissues, the polymorphism of deiodinases must be called into question. In recent studies, when comparing the presurgical hormonal status of patients that had undergone thyroidectomy with their postsurgical status, an association between low FT3 values and Thr92Ala-DIO2 polymorphism was shown even if the patients are formally euthyroid because they are treated with LT4. On the other hand, it is well-known that type 2-idothyronine deiodinase (D2) is largely expressed in mammalian nerve tissue, so its less efficient functioning could lead to a peripheral deficit of T3 and therefore T2, with the effects described above [29]. The Thr92Ala-DIO2 polymorphism involves over 30% of the population.

The effects of this polymorphism on lipid metabolism, leading to an increase in BMI, have also been described. It could therefore be said that these subjects are at risk of systemic or peripheral hypothyroidism [30,31,32]. These findings would once again lead back to the relationship of weight gain with thyroidectomy and may find a better explanation in the metabolic syndrome, a broader view of simple weight gain involving insulin resistance, glucose intolerance or diabetes mellitus, dyslipidemia and hypertension, as well as abdominal obesity.

Zihni (2018) found that despite an effective LT4 replacement, the lipid profile and blood pressure parameters worsened, while the mean body mass index and waist circumference significantly increased at 12 and 24 months of follow-up in 59 patients who underwent total thyroidectomy.

In this study, the author found a trend of T3 reduction towards the lower limits of the normal range [33].

Body weight gain, metabolic syndrome, and chronic asthenia, although different from some pathogenetic stages and clinical aspects that we have already examined, are nosological entities that should be interpreted from different points of view with a unitary approach. There are several publications that demonstrate the convergence of some metabolic pathways involved in each of them.

Several studies have found that these metabolic pathways interact with other substances such as leptins, adiponectins, and melatonin, which are involved in lipid metabolism, insulin resistance, oxidative stress, wake–sleep rhythm, and fatigue resistance.

With regard to melatonin, it does not seem unnecessary to point out that it is also produced extrapineally, primarily by the C cells of the thyroid gland, and it interacts closely with the metabolism of thyroid hormones [34,35,36,37].

Finally, an area that is still unexplored but where some evidence is emerging suggests a possible role for the interrelationship between the intestinal microbiota and l-thyroxine, the absorption and metabolism of which is in turn inversely proportional to gastric acidity [38].

From a therapeutic perspective, the administration of T3 or other nutraceutical drugs such as melatonin or resveratrol, which are powerful activators of mitochondrial metabolism, and, finally, a treatment for intestinal dysbiosis could be evaluated in the future [36,38,39]. From the surgeon’s point of view, total thyroidectomy remains a widely practiced option in most malignant neoplastic forms [40,41], but in light of the most current knowledge, it should be strongly limited, when possible, in benign diseases [15,17].

## 5. Conclusions

The clinician involved in the management of patients undergoing thyroidectomy knows well and handles complications easily. However, he is more often found to have reported vague symptoms that, in any case, could affect well-being and the quality of life. This information should be shared in the team treating patients before and after thyroidectomy, and further investigation of the endocrine-metabolic implications potentially involved should be carried out continuously. 

Until now, there are no proven prevention protocols available, and from the point of view of a possible therapeutic approach, it is possible to formulate only heuristic hypotheses, which are undoubtedly suggestive but without evidence-based confirmation.

At this moment, we can only state that patients who are candidates for thyroidectomy should be adequately informed of both possible implications of the surgical act in terms of weight gain and the onset of chronic asthenia: these conditions are emerging sequelae of thyroidectomy, can have a significant negative impact on the quality of life, and, especially with regard to the effects on body weight, can also constitute a possible comorbidity.

## Figures and Tables

**Figure 1 jcm-11-05486-f001:**
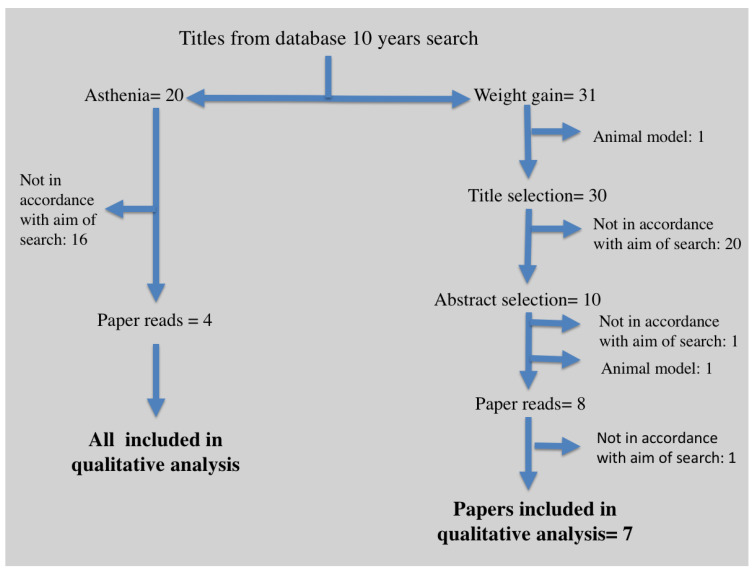
Literature search process and article selection.

**Table 1 jcm-11-05486-t001:** Summary of the characteristics of the studies that concerned weight gain included in the present review. In particular, we highlighted the benchmarks of each study, the most important results in relation to the purpose of this review, the pathogenetic hypotheses that may have been formulated in the study on the relationship between thyroidectomy and weight gain, and finally, the overall opinion formulated on the existence of this relationship (yes/no). Only one study (Park, 2019) expressed this opinion in a doubtful way, articulating it on the coexistence of possible pre-existing risk factors with respect to thyroidectomy, such as alcohol consumption and preoperative obesity. Legend: §: assessed according to Sackett (7). Euthyr, euthyroid; hypothyr, hypothyroid; hyperthyr, hyperthyroid; WG, weight gain; Tx, thyroidectomy.

Author	Year	Patients Enrolled	Control Group(s) (If Any)	Evidence Level §	Benchmark	Results	Pathogenic Hypothesis	Correlation with Tx
Jonklaas	2011	120 euthyr benign	- 120 hypothyr (follow-up) - 120 thyroid cancer LT4 suppressive treatment - 120 healthy subjects	Sackett 4	- WG/BMI Comparison - among groups - Comparison between sexes	- WG/BMI increase: 86% Tx 70% hypothyr more evident in menopause	- Surgical stress - LT4 disfunction - LT3 peripheral ↓ - Physiological/psychosocial	YES
Weinreb	2011	102 thyroid cancer	92 benign (follow-up)	Sackett 4	- Long-term weight and BMI changes	- No changes even excluding suppressed TSH	Adequate LT4 dosage	NO
Rotondi	2014	267 benign/malignant/hyperthyr	Longitudinal evaluation	Sackett 4	Effects of Tx on body weight, excluding any condition other than thyroid function	- WG in 58% - Weight loss in 22% - Stable in 19.5% - regardless of demographics, preoperative TSH, disease, Tx extent	- ↑ body water - Role of leptin in: - energy homeostasis - TRH gene expression conversion T4 → T3 - Changes in lifestyles, - sleep–wake rhythm, - physical activity	YES
Lang	2016	898 benign non-toxic goiter	- 179 non-toxic goiter (follow-up) - 80 parathyroid adenoma	Sackett 3	Baseline: perioperative period/ first outpatient examination. Weight variations 12 months before, 6 + 12 after baseline	WG in 76.8% of patients in study group vs. 60.9% and 55% in control groups. Tx extent not related to WG	- Sudden change in dietary habits - Lifestyle - Change in body metabolism	YES
Glick	2018	79 benign disease	All divided into 3 groups: WG > 2%; Stable Weight; Weight loss	Sackett 4	- Patient’s average with WG - pre- and postoperative difference in Weight - correlation with TSH	18 months later: - 35% WG - 33% stable - 32% weight loss - Non-significant difference in weight - No correlation with TSH	None	NO
Ospina	2018	435 thyroid disease	181: cancer → Tx 226: follow up 28: benign → Tx	Sackett 3	Weight and BMI changes at 1, 2, and 3 years of follow-up	Slight weight gain in all the groups examined	None	NO
Park	2019	227 thyroid disease	38.8%: BMI > 25 61.2%; non-obese	Sackett 4	- Prevalence of post-Tx obesity - Tx extent and obesity	Obesity: 38.8% → 39.6% Tx extent: no influence	Thyroid function → Lipid profiles	Risk factors?

**Table 2 jcm-11-05486-t002:** Summary of the main aspects of the only two articles found in the literature concerning the association between chronic asthenia and thyroidectomy. As in the Table 1, benchmarks, highlights, pathogenic hypotheses, and overall opinion of each study have been reported. Legend: §: assessed according to Sackett (7). WG, weight gain; Tx, thyroidectomy; TL, thyroid lobectomy; TT, total thyroidectomy; B, benign; C, cancer. # in both studies thyroidectomy and thyroid lobectomy were compared; some patients were excluded during the study and therefore not considered in the total number of patients included because the inclusion criteria failed during follow-up.

Author	Year	Patients Enrolled #	Control Group(s) (If Any) #	Evidence Level §	Benchmark	Results	Pathogenic Hypothesis	Correlation with Tx
Rosato	2014	95 uninodular (TL)	96 multinodular/diffuse (Tx)	Sackett IV	Prevalence of asthenia in Tx	Asthenia: Tx = 25%; TL = 0	Inadequate T3 levels	YES
Scerrino	2017	177 suffering from goiter (excluding malignancy/Graves’/severe chronic diseases/hypothyroidism)	54 Uninodular disease	Sackett III	- Impact of asthenia in Tx - Correlation of asthenia with habits (smoking, alcohols), moderate chronic diseases, mental disorders	Increase in asthenia: Tx = 36.16%; TL = anecdotal No correlation of asthenia with habits, chronic diseases, mental disorders	Deficit in “nonclassical” thyroid hormones (T2) that have intense stimulating activity on mitochondrial dehydrogenases and oxygen consumption	YES
Luddy	2021	192 patients: - 67 TL-B - 32 TL-CA - 40 TT-B - 43 TT-CA		Sackett III	Impact of different procedures, for different disease, in appearance of asthenia	Asthenia: - 42% for TT - 35% in TT-B group - 48.8% in TT-CA group - 4% for TL - 3% in TL-B group - 6.25% in TL-CA group	High risk of asthenia in TT group Differences in benign disease/malignancy	YES
Scerrino	2022	233 TT	- 43 ptx°° - 43 cholecystectomy	Sackett III	Differences in asthenia trend between TT and control groups evaluated at baseline, 6 months after surgery, and 12 months after surgery	- TT: significant increase in asthenia from baseline to control periods - Ptx: significant decrease in asthenia from baseline to control periods - Cholecystectomy: asthenia stable in the 3 periods	- The same as Scerrino (2017) - Melatonin? - Gut dysbiosis?	YES

## Data Availability

The data presented in this study are avalaible from the corresponding author upon reasonable request.

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
