# Peer review of "Weight Gain and Asthenia Following Thyroidectomy: Current Knowledge from Literature Review"

_jcm, 2022, doi:10.3390/jcm11185486_

Round 1

Reviewer 1 Report

I have read this manuscript with interest and I think that it is an important contribution to the field. This is a systematic review carried out with the aim of investigating weight gain and the development of asthenia following total thyroidectomy. The work is well written and properly structured. The bibliographic search has been correctly set up according to the PRISMA citeria. The analysis of the selected works is interesting and discussion and conclusions are well argued on the basis of what is reported in the results.

I have only minor concerns:

- The text is well written, however there are some typos, please check the text again (e.g. line 80 a full stop is missing at the end of the sentence...)

- I advise the authors to add two separate sub-chapters in the results section, so as to divide the results' presentation regarding weight gain and asthenia.

- I suggest the Authors to insert tables 1 and 2 at the end of the work or before the discussion.

In conclusion, I believe that the work is valid

Reviewer 2 Report

This paper reviews the current evidence on the association between thyroidectomy and weight change and asthenia. This is an interesting topic focusing on the postoperative quality of life. The authors did an excellent job to summarize the present literature and point out the direction of future investigation.

Please address the following concerns:

The most important parts of this manuscript are Table 1 and Table 2. However, the symbols and abbreviations used in the tables were difficult to read. The mixtures of symbol and abbreviation to represent one item are difficult to understand.

 a)      In Table 1, TxØ = thyroidectomy. The “Ø” seems redundant, the authors may consider Tx=thyroidectomy. Similar scenarios in Table 2, “TLØØ” = thyroid lobectomy. The authors may consider TL=thyroid lobectomy. Also, “Ø” for thyroidectomy, and “Ø Ø” for lobectomy are counter-intuitive. It may be more logical that “Ø” represent one side (lobectomy) and “ØØ” represent two sides (total thyroidectomy)

b)      WG@=weight gain. You may use WG=weight gain and omit the “@”

c)      Euthyr*, thyr*, hyperthyr*. Use “*” (1 character) to replace “oid” (3 characters) is not efficient.

d)      In Table 2, TL-B*, TL-CA**. What is the different between one asterisk and two asterisk?

e)      In Table 2, TT-B+, TT-CA++. The author did not define “+” in the legend and what is the difference between one plus and two plus? Also, the author should add the abbreviation in the table legend (B=benign, CA=cancer).

f)       In table 1 and 2, the level of evidence is assessed by Sackett(9). This is the 7th reference, not the 9th reference.
